# Associations between impulsivity and self-care adherence in individuals diagnosed with Type 2 or prediabetes

Katherine Wainwright[1], Paul Romanowich[2]*, Meghan A. Crabtree[3]

**1** Department of Psychology, University of Texas at San Antonio, San Antonio, Texas, United States of America, **2** Department of Psychology, Gonzaga University, Spokane, Washington, United States of America, **3** Department of Psychology, Colorado State University, Fort Collins, Colorado, United States of America

* romanowich@gonzaga.edu

**Data Availability Statement:** All relevant data are within the paper and its Supporting Information files.

**Funding:** The authors received no specific funding for this work.

## Abstract

Diabetes is a chronic disease requiring extensive self-care. Different impulsivity constructs, including choice-based and self-report personality measures are associated with decreasing diabetes self-care adherence. However, both choice-based and self-report impulsivity have never been measured for individuals diagnosed with either Type 2 or prediabetes in the same study. The current study examined the relationship between impulsivity and diabetes self-care in 101 adults diagnosed with either Type 2 or prediabetes. Results indicated that increasing self-reported impulsiveness was significantly correlated with decreasing Type 2 diabetic self-care, whereas the choice-based measure was not associated with any self-care measure. No association between impulsivity and self-care was significant for individuals diagnosed with prediabetes. Path analyses showed that self-reported impulsiveness directly and positively predicted problems controlling blood sugar levels in individuals diagnosed with either prediabetes or Type 2 diabetes. However, self-reported impulsiveness only indirectly and negatively predicted exercise and diet adherence via diabetes management self-efficacy for individuals diagnosed with Type 2 diabetes. These results show what specific impulsivity constructs and diabetes management self-efficacy may be incorporated into interventions for increasing specific self-care behaviors.

## Introduction

Diabetes is a chronic condition, impacting an estimated 422 million individuals worldwide (~17% global population), whose pancreas either cannot produce any or enough insulin leading to multiple health complications such as hypoglycemia, eye problems, limb amputations, hypertension, increased heart attack and stroke risk, and kidney disease [1]. Type 1 diabetes is congenital and accounts for about 5% of all diabetes diagnoses, whereas Type 2 diabetes is not congenital and accounts for > 90% of all diabetes diagnoses [1]. Prediabetes, where blood sugar levels are elevated for an extended time [2] is a Type 2 diabetes precursor [3] and impacts an even larger number of individuals (33.9% of US adults; [2]). Both Type 2 and prediabetes,

**Competing interests:** The authors have declared that no competing interests exist.

are largely preventable through a balanced diet and periodic exercise [1]. Thus, behavioral interventions (e.g., blood glucose monitoring, diet and exercise adherence) are important for reducing and maintaining normal blood sugar levels [4], in addition to pharmacological treatments [1]. The current study focused on individuals diagnosed with Type 2 and prediabetes due to the high prevalence rates and preventability described above. However, self-care plan complexity (e.g., changing diet, increasing or starting exercise, medication regiments), can result in low treatment adherence rates [5]. That is, both behavioral and pharmacological interventions contain adherence requirements that can decrease compliance [5].

Self-efficacy, an individual's judgment about their capabilities to organize and execute actions, is a significant predictor for diabetes self-care adherence [4, 6]. Those individuals that self-report higher diabetes management self-efficacy (e.g., to what extend do you feel able to choose the correct foods) are more likely to adhere to behavioral and pharmacological interventions [4, 6]. However, difficulty following through with self-care behaviors may be multiply determined. For example, different impulsivity constructs have been linked to multiple detrimental health behaviors (e.g., obesity, substance abuse, [7, 8]). One aspect of impulsivity, delay discounting (DD), is a behavior gratification measure and involves a series of choices used to assess an individual's preference between smaller-sooner and larger-later options [9]. For diabetes, increased DD may result in individuals engaging in behaviors that have short-term reinforcement properties (e.g., drinking products high in sugar), at the expense of long-term treatment and management goals (i.e., reducing blood sugar levels). As described below, other aspects of impulsivity, such as a lack of inhibition, also have been shown to predict diabetes self-care adherence. However, a lack of inhibition (as measured by the Barratt Impulsiveness Scale-11 [BIS-11]), typically does not significantly correlate with delay of gratification (as measured by DD; [10]). Thus, these impulsivity constructs will be introduced separately in regard to their association with self-efficacy and diabetes self-care adherence.

One group of researchers has examined the relationship between DD and Type 2 diabetes self-management. Reach and colleagues [11] found that increased DD was associated with non-adherence to medication and increased $HbA1_c$ (average glycated hemoglobin–a biologic marker for increased blood sugar) levels in adults. Higher $HbA1_c$ levels typically reflect worse diabetes management. During a follow-up, Lebeau et al. [12] used a DD task with delays ranging from 3 days to 10 years. Results continued to show a positive association between DD and $HbA1_c$ levels. Medication adherence partially mediated the relationship between DD and $HbA1_c$ levels. However, they found no relationship between DD rates and diet adherence.

Another study examined the relationship between DD and non-adherence and poor glycemic control within a sample of adolescents with Type 1 diabetes [13]. DD was measured using a computerized procedure with delays of 1 day, 1 week, 1 month, and 6 months and a larger delay reward of $1000. Lansing et al. [13] found that increased DD was associated with worse diabetes self-care in adolescents. The relationship between DD and glycemic management was moderated by direct parent observation of care, suggesting a potential mode of intervention. DD was not related to self-monitoring frequency for blood glucose levels.

More recently researchers measured the relationship between glycemic control, medication adherence and DD in adults diagnosed with prediabetes [14]. Like the results for individuals diagnosed with Type 2 diabetes [11, 12], individuals diagnosed with prediabetes showed positive correlations between DD rates and $HbA1_c$ levels. Additionally, there were significant negative relationships between DD rates and medication adherence, diet quality and physical activity. Thus, researchers have consistently found a positive relationship between DD rates and glycemic management across different diabetes diagnoses (Type 1, Type 2, prediabetes) and ages (adolescents and adults). In sum, the results from these studies suggest that impulsivity measured via the DD task are associated with poorer diabetes self-care adherence. However,

the findings are less consistent for behaviors related to glycemic control (e.g., diet adherence) for adults.

The BIS-11 [15] is frequently used as a self-report measure of the inhibition construct for impulsivity and is comprised of three subscales: Attentional, Motor, and Nonplanning. Theoretically, all three subscales should be related to diabetes self-care, as adhering to a diabetes treatment regimen requires attention to detail, forethought, and planning. One study examined the relationship between the BIS-11 and body mass index (BMI) in a sample diagnosed with Type 2 diabetes. Raymond and Lovell [16] found that the BIS-11 was positively related to BMI, with food addiction and nonplanning impulsiveness accounting for 38% of the variance in BMI. Although this study did not assess diabetes self-care adherence behavior or glycemic management, the results provide evidence that other impulsiveness measures are important predictors for BMI among adults diagnosed with Type 2 diabetes. Elevated BMI is a risk factor for Type 2 and prediabetes diagnoses [17], making a relationship between impulsiveness and BMI important to consider.

More recently, Hadj-Abo et al. [18] assessed the relationship between self-reported HbA1c levels, diabetes self-management, impulsivity as measure through the BIS-11, diabetes-specific self-efficacy, and need for cognition in individuals diagnosed with Type 2 diabetes. Like the DD studies described above [11, 12, 14], self-reported impulsivity was significantly negatively associated with self-reported HbA1c levels. Diabetes self-management was also significantly negatively related to BIS-11 score; as impulsivity increased, diabetes self-management decreased. An additional mediation analysis showed that self-efficacy fully mediated this relationship, whereby the effect of impulsivity on diabetes self-management was dependent on self-efficacy. This result was like other important health behaviors that have also measured self-efficacy and self-management [4, 6]. However, unlike previous studies, diabetes self-care management was not broken down into the component subscales of self-care (e.g., diet, physical activity, blood sugar control, and foot care) and analyzed separately. This is important as previous studies have shown differential relationships between impulsivity and self-care management [12–14].

Given the large number of individuals diagnosed with diabetes [1] and prediabetes [3], identifying behavioral processes that influence adherence has potential for improving health and preventing Type 2 diabetes in individuals already diagnosed with prediabetes. No study focusing on impulsivity and self-care behavior has included individuals diagnosed with prediabetes and used different impulsivity measures in the same sample. Previous studies have demonstrated that DD and the BIS-11 measure different impulsivity constructs [10, 19, 20]. Therefore, if each impulsivity construct (i.e., DD and BIS-11) is differentially related to self-care adherence in these populations, then only that impulsivity construct would be an additional target for an intervention to improve or reverse negative health outcomes.

From the previously described literature, it was hypothesized that individuals with higher impulsivity will have fewer numbers of days adhering to self-care activities and more management problems. Higher self-efficacy would be associated with increased self-care and management understanding, as well as fewer management problems and perceived barriers to care. Self-efficacy would mediate the relationship between impulsivity and diabetes self-care, similar to previous research [18]. Lastly, differences in impulsivity (DD and BIS-11), self-efficacy, and self-care between individuals diagnosed with Type 2 diabetes compared to those diagnosed with prediabetes were also tested for. Given the lack of research comparing these two groups, the direction of these differences was not specified a priori.

## Materials and methods

### Participants

With Institutional Review Board approval from the University of Texas, San Antonio, 110 participants over the age of 30 years who self-reported a diagnosis of either Type 2 diabetes or prediabetes were recruited through Qualtrics (an internet-based survey company) and incentivized through their preferred method (e.g., airline miles, gift cards). Participants over 30 were recruited based on the timelines to develop both Type 2 and prediabetes. That is, it typically takes years of poor diet and sedentary behavior for Type 2 or prediabetes to develop [1]. Informed consent was obtained online for all participants. In this case, participants consented to study involvement by clicking "I accept" to the study protocol outlined on the consent webpage. There was no penalty for not consenting. However, potential participants who did not click "I accept" were redirected back to a landing page. Self-reported diagnoses were mutually exclusive. Nine participants were excluded from analysis for inconsistent responding. Statistical analysis included 101 participants. These 101 participants successfully completed all measures. Therefore, there was no missing data.

Participants' age ranged between 30 and 84 years (M = 51.03, SD = 12.58) and 53% were female (Table 1). Fifty participants self-reported having a diagnosis of Type 2 diabetes with an average of 9.04 years (SD = 7.4 years) since diagnosis and 51 self-reported being diagnosed with prediabetes with an average of 3.37 years (SD = 3.83). BMI ranged from 17.79 to 65.73 (M = 33.03, SD = 8.73).

**Table 1. Descriptive demographic data.**

|  |  | Type 2 | Prediabetes |
|---|---|---|---|
|  | N = | 50 | 51 |
| Gender | Males (%) | 19 (38) | 29 (57) |
|  | Females (%) | 31 (62) | 21 (43) |
| Ethnicity | White/European (%) | 40 (80) | 41 (80) |
|  | African American/Black (%) | 4 (8) | 3 (6) |
|  | Asian American (%) | 2 (4) | 4 (2) |
|  | Hispanic/ Latino American (%) | 3 (6) | 3 (6) |
|  | Other (%) | 1 (2) | 2 (4) |
| Education | Some high school (%) | 1 (2) | 0 |
|  | High school graduate/GED (%) | 11 (22) | 5 (10) |
|  | Some college, no degree (%) | 10 (20) | 15 (29) |
|  | Trade/technical training (%) | 2 (4) | 2 (4) |
|  | Associate's degree (%) | 9 (18) | 6 (12) |
|  | Bachelor's degree (%) | 12 (24) | 16 (31) |
|  | Master's degree or higher (%) | 5 (10) | 7 (14) |
| Employment | Full time (%) | 22 (44) | 27 (53) |
|  | Part time (%) | 4 (8) | 7 (14) |
|  | Retired (%) | 16 (32) | 9 (18) |
|  | Other (%) | 8 (16) | 8 (16) |
| Income (USD) | < $19,999 (%) | 6 (12) | 4 (8) |
|  | $20,000 - $29,999 (%) | 7 (14) | 9 (18) |
|  | $30,000 - $39,999 (%) | 7 (14) | 8 (16) |
|  | $40,000 - $49,999 (%) | 9 (18) | 3 (6) |
|  | $50,000 - $59,999 (%) | 3 (6) | 7 (14) |
|  | $60,000 + (%) | 18 (36) | 20 (39) |

## Measures

All measures were normally distributed unless otherwise noted.

**Management & adherence.** Summary of Diabetes Self-Care Activities (SDSCA; [21]). The SDSCA includes 20 questions assessing self-care levels over the past 7 days related to diet, exercise, blood sugar testing, foot care, and medication. For example, participants were asked on a scale from 0–7 to indicate on how many of the past seven days they followed a healthful eating plan. Higher scores indicated more days of adherence. The Diet ($\alpha = 0.74$) and Exercise ($\alpha = 0.86$) subscales demonstrated adequate reliability. Previous studies have shown that internal consistency measured via inter-item correlations is acceptable and correlations with other standardized diet and exercise measures are significant [22]. In the current study, the Foot Care scale for individuals with Type 2 diabetes did not have adequate reliability ($\alpha = 0.24$) and was excluded from analysis. Medication and blood sugar testing were composed of individual items.

Diabetes Care Profile (DCP; [23]). Three subscales from the DCP were used: 1) The Control Problem scale ($\alpha = 0.94$) contained 19 items that assess instances of poor diabetes management and potential reasons why blood sugar levels were not managed. Responses for all three subscales were on a 1–5 scale with participants indicating how often an event occurred in the past year. Higher scores indicated more difficulty managing the condition. The Control Problem scale was not normally distributed. Transformations did not correct the distribution and nonparametric statistical methods were used for this scale, 2) The Barriers to Testing scale was presented to participants who indicated regularly testing their urine and/or blood for sugar. The scale contained 11 items ($\alpha = 0.88$) assessing how frequently barriers impede their ability to test (e.g., cost of testing). Due to low event occurrence (i.e., positive skew) the Barriers to Testing scale was normalized using a natural logarithm function. 3) The Understanding Scale, answered only by participants with Type 2 diabetes, contained 13 questions ($\alpha = 0.94$) assessing understanding of diabetes related issues such as foot care. Higher scores for each scale indicated more difficulty managing, more barriers, and a greater understanding of diabetes, respectively. Previous research has shown similarly high Cronbach's alphas in English-speaking populations [22–24].

**Self-Efficacy.** The Self-Efficacy for Diabetes (SED; [25]) is an 8-item questionnaire ($\alpha = 0.92$) that assesses an individual's confidence completing diabetes management practices (e.g., exercise). Participants responded on a 10-point Likert scale from "not at all confident" to "totally confident". Higher scores indicated greater self-efficacy. Previous data analyses indicated good internal consistency for both English ($\alpha = 0.83$) and Spanish-speaking ($\alpha = 0.85$) patients diagnosed with Type 2 diabetes [25].

**Impulsiveness.** The Barratt Impulsiveness Scale-11 (BIS-11, [15]) asks participants to answer questions about how often they engaged in each of 30 statements describing impulsive or non-impulsive behaviors to create a total impulsiveness score across three second-order factors; Attentional (8 items; $\alpha = 0.74$), Motor (11 items; $\alpha = 0.75$), and Nonplanning (11 items; $\alpha = 0.74$). Participants were asked to use a 4-point response option scale of 'never' to 'always' to indicate how often they engaged in the behavior described in each statement. Some items were reverse scored. Higher scores indicated more impulsiveness. Cronbach's alphas between 0.79 and 0.83 have been reported in the general population [15]. A Cronbach's alpha = 0.95 was recently reported for individuals diagnosed with Type 2 diabetes [18].

**Delay discounting.** The delay discounting task (DD, [8]) consisted of 27 items asking individuals to make a choice between a small immediate reward and a larger but delayed reward (e.g., $25 right now or $60 in 14 days). A free parameter $k$ indicated discounting curve steepness which corresponded with the geometric midpoint of the ranges [26]. Higher $k$-values

indicated more discounting. $k$-values were normalized using a natural logarithm function. Response consistency scores ($R^2$) ranged from 0.81 to 1 (M = 0.96, SD = 0.04). A within subjects ANOVA with a Greenhouse-Geisser correction provided evidence of a magnitude effect ($F(1.82, 178.94) = 18.04$, $p < 0.001$), supporting the criterion validity of the measure within this sample. Previous studies have shown the DD task to be highly consistent and valid [26].

## Analysis

Pearson zero-order correlations were used to determine if increased scores on the DD task and the BIS-11 (i.e., more impulsiveness) were associated with fewer numbers of days adhering to self-care activities and more management problems. Correlations were used to examine the association between self-efficacy to self-care outcomes, barriers, and understanding of management issues. A series of independent sample $t$-tests were used to compare individuals diagnosed with Type 2 diabetes compared to those diagnosed with prediabetes in DD, BIS-11, self-efficacy, and self-care. Mann-Whitney U tests were performed on variables that were not normally distributed. All correlations and statistical tests were conducted using SPSS (Version 25).

Finally, two fully saturated path models were specified to estimate the direct and indirect effects of DD and BIS-11 on self-care outcomes via diabetes management self-efficacy for individuals diagnosed with either Type 2 diabetes or prediabetes. Previous psychometric evaluations for the BIS-11 suggest that a single factor structure provides a better fit to the data [27, 28]. Therefore, the total BIS-11 score was used in both path analyses. Path analyses were conducted with maximum likelihood estimation. Indirect effects were estimated using bias-corrected bootstrapped 95% confidence intervals ($n = 1000$), per [29]. Path models were tested using MPlus.

## Results

### Type 2 diabetes vs. prediabetes

Measurement means, standard deviations, and ranges are presented in Table 2. Independent sample $t$-tests were used to compare individuals diagnosed with Type 2 diabetes to those

**Table 2. Means (SD) and ranges for each measure by diagnosis.**

| Measure | Type 2 Diabetes | | Prediabetes | | Independent sample $t$-tests |
|---|---|---|---|---|---|
| | Mean (SD) | Range | Mean (SD) | Range | $t(99) =$ |
| SDSCA Diet | 4.73 (1.46) | 1–7 | 4.05 (1.75) | 0–7 | 2.09* |
| SDSCA Exercise | 3.51 (2.15) | 0–7 | 3.30 (2.27) | 0–7 | 0.468 |
| DCP Understanding | 4.10 (0.73) | 2.75–5 | N/A+ | N/A+ | N/A+ |
| DCP Control | 2.04 (0.84) | 1–4.05 | 1.98 (0.79) | 1–4.74 | $U = 1003$ |
| DCP Barriers# | 0.22 (0.19) | 0.00–0.65 | 0.30 (0.18) | 0.00–0.68 | -1.65 |
| SED | 7.76 (2.97) | 2–10 | 6.43 (2.17) | 2–10 | 2.88* |
| BIS-11 Attention | 16.74 (4.48) | 9–31 | 15.47 (4.57) | 8–26 | 1.41 |
| BIS-11 Motor | 21.70 (4.97) | 14–39 | 22.63 (5.12) | 13–40 | -0.92 |
| BIS-11 Nonplanning | 23.86 (5.49) | 13–35 | 23.29 (5.12) | 13–37 | 0.53 |
| BIS-11 Total | 62.30 (11.19) | 39–88 | 61.39 (12.14) | 37–89 | 0.39 |
| Delay Discounting# | -1.71 (0.71) | -3.8 - -0.60 | -1.59 (0.72) | -3.8 - -0.60 | -0.71 |

#Transformed variable.

* $p < 0.05$.

+DCP Understanding was not measured for individuals with prediabetes, as they would not yet be encountering some of the symptoms described (e.g., foot care).

diagnosed with prediabetes, except for DCP Control, which was tested with a Mann-Whitney U test. There were no significant differences for DD, any of the BIS-11 subscales or exercise. Individuals diagnosed with Type 2 diabetes reported higher self-efficacy (M = 7.76, SD = 2.97) compared to individuals diagnosed with prediabetes (M = 6.43, SD = 2.17, $t(99)$ = 2.88, $p < 0.01$). This difference had a medium effect size ($d$ = 0.51). Individuals diagnosed with Type 2 diabetes also reported more days adhering to a diet (M = 4.73, SD = 1.46) compared to individuals diagnosed with prediabetes (M = 4.05, SD = 1.75, $t(99)$ = 2.09, $p$ = 0.039). Like self-efficacy, the effect size for this difference was medium ($d$ = 0.43). A Mann-Whitney U test showed no significant differences between groups on the control scale.

## Between and within scale associations

As shown in Table 3, there were no significant Pearson zero-order correlations between DD and any of the three BIS-11 subscales for either individuals diagnosed with Type 2 or prediabetes, replicating previous null results (e.g., [30]) and suggesting that each measure different impulsivity constructs. However, for individuals diagnosed with prediabetes there was a significant positive correlation between DD and total BIS score. All BIS-11 subscales positively correlated with each other (all $r$'s > 0.374, $p < 0.05$), except for the BIS Motor and Nonplanning subscales for individuals diagnosed with Type 2 diabetes. Table 3 also shows the relationships between impulsivity and adhering to self-care activities and diabetes management, via correlations between all subscales calculated separately for each diagnosis.

For all individuals, DD was not significantly correlated to any self-care or management problems or self-efficacy, as shown in the leftmost column of Table 3. For individuals

**Table 3. Pearson zero-order correlations between impulsivity, self-efficacy, and self-care.**

| | | DD | BIS attn. | BIS motor | BIS nonplan | BIS total | SED |
|---|---|---|---|---|---|---|---|
| Type 2 | SDSCA Blood | -0.053 | 0.027 | 0.207 | -0.264 | -0.004 | 0.307* |
| | SDSCA Medication | -0.054 | -0.241 | -0.263 | -0.294* | -0.238 | 0.022 |
| | SDSCA Diet | -0.069 | -0.306* | -0.011 | -0.393* | -0.320* | 0.646* |
| | SDSCA Exercise | 0.059 | -0.188 | 0.226 | -0.439* | -0.190 | 0.742* |
| | DCP Understanding | -0.136 | 0.058 | 0.004 | -0.387* | -0.165 | 0.533* |
| | DCP Control | -0.066 | 0.306* | 0.339 | 0.246 | 0.411* | -0.343* |
| | DCP Barriers[+] | 0.087 | 0.092 | 0.252 | 0.342* | 0.304 | -0.410* |
| | BIS Attention | -0.077 | | | | | |
| | BIS Motor | -0.002 | 0.587* | | | | |
| | BIS Nonplanning | -0.155 | 0.374* | 0.105 | | | |
| | BIS Total | -0.108 | 0.845* | 0.731* | 0.687* | | |
| | SED | 0.063 | -0.252 | 0.032 | -0.631* | -0.396* | |
| Prediabetes | SDSCA Diet | -0.133 | -0.149 | -0.049 | -0.265 | -0.191 | 0.410* |
| | SDSCA Exercise | -0.042 | -0.274 | -0.061 | -0.260 | -0.241 | 0.425* |
| | DCP Control | 0.052 | 0.367* | 0.124 | 0.321* | 0.327* | -0.125 |
| | DCP Barriers[+] | 0.345 | 0.315 | 0.336 | 0.033 | 0.317 | 0.229 |
| | BIS Attention | 0.239 | | | | | |
| | BIS Motor | 0.329* | 0.554* | | | | |
| | BIS Nonplanning | 0.116 | 0.555* | 0.385* | | | |
| | BIS Total | 0.279* | 0.849* | 0.796* | 0.801* | | |
| | SED | -0.057 | 0.014 | 0.128 | -0.179 | -0.018 | |

[+]DCP Barriers scale was presented to participants who tested their urine or blood at least 1 day a week (Type 2: n = 39, Prediabetes: n = 24).

* p < 0.05.

diagnosed with Type 2 diabetes, there was no significant correlation between impulsivity and adherence to medication or blood sugar testing as measured by the SDSCA, with the exception of the BIS-11 Nonplanning scale, where higher scores were associated with less days adhering to medication recommendations. BIS-11 Attention was negatively related to SDSCA Diet and positively related to DCP Control, with higher attention impulsiveness associated with fewer days adhering to diet and increased difficulty managing diabetes. Higher BIS-11 Nonplanning scores were associated with fewer days adhering to both a diet and an exercise regime (SDSCA Exercise), a worse understanding of diabetes (DCP Understanding), and increased perceived barriers to care (DCP Barriers). Increased self-efficacy was associated with better DCP Understanding, increased blood sugar testing, diet, and exercise adherence (measured via the DCP), and fewer management problems and perceived barriers to care (measured with DCP Control and Barriers, respectively). Self-efficacy was negatively correlated with the BIS Nonplanning subscale. However, this was the only significant correlation between self-efficacy and any of the impulsivity measures.

For individuals diagnosed with prediabetes, increased self-efficacy was associated with increased diet and exercise adherence (via SDSCA), but not with any of the impulsivity measures. For individuals diagnosed with prediabetes, higher BIS-11 Attention scores were associated with increased management difficulty, as measured by the DCP Control scale. Lastly, higher BIS-11 Nonplanning scores were also associated with increased management difficulty, as measured by the DCP Control scale.

## Path models

Based on the differences between individuals diagnosed with Type 2 diabetes compared to prediabetes, two path models were specified to estimate the direct and indirect effects BIS-11 on exercise adherence, diet adherence, and diabetes control problems via diabetes management self-efficacy (Figs 1 and 2). As DD was uncorrelated with all other study variables for individuals with either prediabetes or Type 2 diabetes, the variable was excluded from the path analyses. Given the relationship between impulsivity and BMI/obesity [31], and the higher rates of BMI in populations with diabetes [2], BMI was specified as a covariate within both models. With BMI excluded as a covariate two relationships changed, as specified in the prediabetes group below.

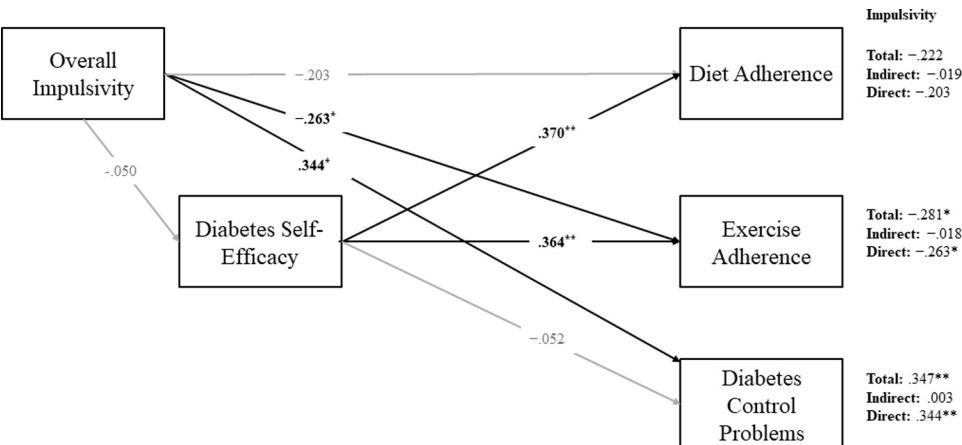

**Fig 1. Path model estimating direct and indirect effects of impulsivity on self-care outcomes among individuals diagnosed with prediabetes, controlling for BMI.** Completely standardized maximum likelihood parameter estimates shown. Unstandardized slopes of the indirect effects along with the bias-corrected 95% bootstrapped confidence intervals are provided in text. $^*p < .05$, $^{**}p < .01$.

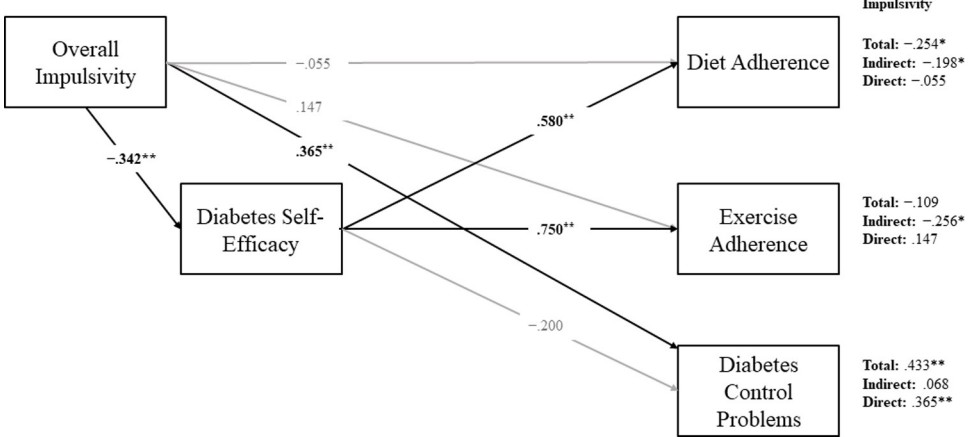

**Fig 2. Path model estimating direct and indirect effects of impulsivity on self-care outcomes among individuals diagnosed with Type 2 diabetes, controlling for BMI.** Completely standardized maximum likelihood parameter estimates shown. Unstandardized slopes of the indirect effects along with the bias-corrected 95% bootstrapped confidence intervals are provided in text. $^*p < .05$, $^{**}p < .01$.

**Prediabetes.**　Among individuals diagnosed with prediabetes, BMI, BIS-11 and self-efficacy accounted for a significant proportion of variability in exercise adherence ($R^2 = .282$, $p = .008$) and a significant proportion of variability in diet adherence ($R^2 = .221$, $p = .031$). The model predictors failed to explain significant variability in control problems ($R^2 = .138$, $p = .149$).

BMI was significantly, negatively related to self-efficacy ($\beta = -.258$, SE = .026, $p = .035$), but was uncorrelated with both the BIS-11 and all three self-care outcomes. Total BIS-11 was not significantly related to self-efficacy. Self-efficacy was positively associated with both diet ($\beta = .370$, $SE = .151$, $p = .025$) and exercise adherence ($\beta = .363$, $SE = .170$, $p = .011$) but unrelated to control problems. Total BIS-11 had no direct relation to diet adherence. However, total BIS-11 had a direct, positive relation to control problems ($\beta = .344$, $SE = .128$, $p = .007$) and a significant negative relationship to exercise adherence ($\beta = -.263$, $SE = .026$, $p = .026$). When BMI was excluded as a covariate the relationships between total BIS-11 and control problems ($\beta = .33$, $p = .068$) and exercise adherence ($\beta = -.23$, $p = .087$) were no longer significant.

Total BIS-11 and self-efficacy were both significantly, directly related to the self-care outcomes summarized above in expected directions. Total BIS-11 was directly, positively related to control problems and negatively related to exercise adherence, while self-efficacy positively related to both diet and exercise adherence. Self-efficacy did not mediate the relationship between BIS-11 and self-care outcomes among individuals diagnosed with prediabetes.

**Type 2 diabetes.**　Among individuals diagnosed with Type 2 diabetes, BMI, BIS-11 and self-efficacy explained a significant and relatively large proportion of variability in exercise ($R^2 = .587$, $p < .001$) and diet adherence ($R^2 = .441$, $p < .001$). Model predictors explained a significant but small proportion of variability in control problems ($R^2 = .208$, $p = .048$).

BMI was not related to exercise ($\beta = -.165$, SE = .097, $p = .088$), self-efficacy ($\beta = -.223$, SE = .127, $p = .080$), and uncorrelated with all other variables. Total BIS-11 was significantly negatively associated with self-efficacy ($\beta = -.342$, SE = 0.123, $p = .006$). Self-efficacy was positively associated with both diet ($\beta = .580$, $SE = .102$, $p < .001$) and exercise adherence ($\beta = .750$, $SE = .078$, $p < .001$). There was no effect of self-efficacy on control problems. However, total BIS-11 was directly positively related to control problems ($\beta = .365$, $SE = .133$, $p = .006$). There were significant negative indirect effects of total BIS-11 via self-efficacy on both diet ($\beta = -.198$, b =

−.026, *95% CI*: [−0.060, −0.006]) and exercise adherence ($\beta$ = −.256, b = −.049, *95% CI*: [−0.112, −0.007]). The direct effect of total BIS-11 on diet and exercise adherence was not significant after accounting for this indirect effect.

Total BIS-11 and self-efficacy were associated with self-care outcomes in the expected directions, showing a direct, positive effect on problems, and an indirect, negative effect on exercise and diet adherence via self-efficacy. Self-efficacy mediated the relationship between BIS-11 and self-care outcomes among individuals diagnosed with Type 2 diabetes.

## Discussion

The present study tested associations between different impulsivity constructs, self-efficacy and self-care behaviors for participants with either self-reported prediabetes or Type 2 diabetes diagnoses. Three main hypotheses were test. First, individuals with higher impulsivity would have fewer numbers of days adhering to self-care activities and more management problem, Second, higher self-efficacy would be associated with increased self-care and management understanding, as well as fewer management problems and perceived barriers to care. Third, self-efficacy would mediate the relationship between impulsivity and diabetes self-care. Differences in impulsivity (DD and BIS-11), self-efficacy, and self-care between individuals diagnosed with Type 2 diabetes compared to those diagnosed with prediabetes were also tested for.

BIS-11 was significantly related to self-care behaviors for individuals diagnosed with Type 2 diabetes, with higher BIS-11 scores associated with fewer days adhering to diet and exercise and increasing control problems. This result was partially consistent with the first hypothesis, as DD was not significantly related to self-care behaviors. The relationship between diabetes self-care and impulsivity was mediated by self-efficacy (see Fig 2) for both diet and exercise adherence. This result replicated the results of Hadj-Abo et al. [18] and was also consistent with the third hypothesis, which predicted that self-efficacy would mediate the relationship between impulsivity and self-care behaviors. The methodology for measuring self-care behaviors was different from Hadj-Abo et al. [18] who used an aggregate self-care management measure in their mediation analyses. Therefore, the current findings suggest not all self-care behaviors are related to impulsivity measures via the BIS-11 for individuals diagnosed with Type 2 diabetes. In addition to self-care, diabetes control problems were also measured via the DCP. Interestingly, for individuals diagnosed with Type 2 diabetes the significant positive relationship between total impulsivity and control problems was not mediated by self-efficacy. There may be limits to the mediating effect of self-efficacy on problems associated with impulsivity and control problems for individuals diagnosed with Type 2 diabetes. More specifically, scales like the DCP questionnaire that rely on an individual's beliefs as to why their diabetes management occurs at a certain level may be a less reliable predictor of actual diabetes management, and thus, unrelated to self-efficacy [32].

Previous studies measuring impulsivity, self-efficacy and self-care behavior have not included individuals diagnosed with prediabetes. The current study aimed to address this gap in the literature. For individuals diagnosed with prediabetes, higher BIS-11 attention and Non-planning scores were associated with increasing management problems. Several other relationships between the BIS-11 and self-care were non-significant but showed a similar pattern to individuals with Type 2 diabetes; higher BIS-11 scores were associated with fewer days adhering to diet and exercise. These results suggest that stronger associations between BIS-11 and diet and exercise adherence may be meaningful markers for individuals transitioning from prediabetes to Type 2 diabetes diagnoses. Perhaps most importantly, self-efficacy was a significant predictor for diet and exercise, whereas BIS-11 was a significant predictor for both exercise and diabetes control problems when BMI was included as a covariate for individuals self-

reporting a prediabetes diagnosis. These results partially supported the hypothesis that self-efficacy would be associated with diabetes understanding, management problems, and perceived barriers. Unlike individuals diagnosed with Type 2 diabetes, self-efficacy did not mediate relationships between impulsivity and any of the self-care behaviors for individuals diagnosed with prediabetes. It may be that diabetes specific self-efficacy for individuals diagnosed with prediabetes is not strong enough to impact these important self-care behaviors, assuming higher self-efficacy causes better self-care adherence. Table 2 shows that self-efficacy is on average more than one point (out of 10) lower for individuals diagnosed with prediabetes, relative to those diagnosed with Type 2 diabetes. It may be that individuals diagnosed with Type 2 diabetes have more experience with self-care behaviors, relative to those individuals diagnosed with prediabetes. As described below, this increased experience with self-care and other factors suggests pathways for important clinical interventions.

This was the first study to measure associations between impulsivity via DD and the BIS-11, and self-care behaviors in individuals diagnosed with prediabetes using an internet-based sample. Although individuals diagnosed with prediabetes were equally impulsive on both measures relative to those diagnosed with Type 2 diabetes, the associations with self-care behaviors were not significant. Prediabetes is not typically associated with noticeable symptoms [3] and individuals have few, if any discriminable signals from their bodies for prediabetes onset. Conversely, many (but not all) individuals diagnosed with Type 2 diabetes typically have noticeable symptoms [1], which may increase motivation to seek a diagnosis and adhere to self-care recommendations. In the current sample, individuals diagnosed with Type 2 diabetes were more likely to adhere to their diet and had higher self-efficacy than individuals diagnosed with prediabetes (see Table 2). Additionally, participants diagnosed with Type 2 diabetes had self-reported being diagnosed almost 3 times as long as those participants with prediabetes. Thus, increased diabetes treatment experience may have contributed to these stronger associations. Future studies using longitudinal designs will be necessary to determine how increasing treatment experience may affect self-efficacy, and/or self-care adherence.

DD was not correlated with self-care behavior regardless of diagnosis nor was it correlated with management problems. The current results failed to replicate the negative relationship between DD and self-care adherence found by Lansing et al. [13] and Reach et al. [11]. However, results replicated Lebeau et al. [12], who also found no relationship between DD and diet. There are a few reasons why the current results did not show a relationship between DD and self-care. First, unlike Reach et al. [11] and Lebeau et al. [12], the main dependent measure for diabetes control was based on self-reports, whereas they obtained quantitative HbA1$_c$ levels. Obtaining HbA1$_c$ levels for an internet-based sample is logistically difficult. DD is a behavioral measure of impulsivity [9] which previous researchers have shown to be either weakly or completely unrelated to self-report impulsivity measures (e.g., [20]). This also suggests that they measure different impulsivity constructs [19]. The diabetes self-care and management measures were self-reports. This might explain why relationships between the BIS-11 and self-care subscales were found in the current study and Hadj-Abo et al. [18], but not with DD and self-care subscales. It may also explain why Lebeau et al. [12] did find a correlation between DD and HbA1$_c$ levels. Self-report measures rely on an individual's ability to correctly perceive, remember, and report their behavior. Future studies should use more objective measures of diabetes self-care and management, such as HbA1$_c$ levels in combination with both DD and self-report impulsivity measures.

Second, treatment regimens for prediabetes and Type 2 diabetes vary considerably for each individual. This variability is dictated by symptomology and demographic variables (e.g., age and family history of Type 2 diabetes). To obtain more precise associations between self-care adherence and psychological constructs like impulsivity and self-efficacy, future studies should

obtain prescribed treatment regimes. This would include any changes to treatment based on changes in HbA1$_c$ levels. That is, more dynamic models could be created incorporating changes in both disease progression and psychological constructs.

Lastly, the DD measure used in this and other research with individuals diagnosed with diabetes has only focused on the value of money [11–13]. Different commodities have been previously tested within the DD paradigm (e.g., food, sex and health). A recent meta-analysis demonstrated that this commodity effect phenomenon was highly reliable for DD [33]. However, there are also pronounced state-like effects for DD, where non-monetary outcomes are discounted at higher rates than monetary outcomes [34]. It may be that discounting for more relevant commodities (e.g., sugary foods for individuals diagnosed with Type 2 diabetes) may show stronger associations with self-care behaviors related to diet (SDSCA diet), relative to monetary outcomes. Likewise, DD for health outcomes may show a stronger association for DCP understanding, relative to monetary or sugary foods as an outcome.

Consistent with Mohebi et al. [4], having a higher level of self-efficacy was associated with better adherence to diet and exercise for all individuals. For individuals diagnosed with Type 2 diabetes, self-efficacy was correlated with a better understanding of management, fewer management problems, and fewer perceived barriers. This is consistent with research that has shown a positive relationship between behavior change and self-efficacy over a range of health-related behaviors [35], though the causal status of self-efficacy has yet to be empirically demonstrated (e.g., [36]). Although the current study cannot provide evidence either for or against the causal status of self-efficacy, it does demonstrate the consistent significant association self-efficacy has with important self-care behaviors for individuals diagnosed with diabetes. Increasing individual's self-efficacy may be one mechanism for a clinical intervention using the current and previous [4, 18] results. However, it should be noted that either manipulating self-efficacy or impulsivity is an indirect route for behavior change. A more direct route would be to directly target those deficient self-care behaviors (e.g., diet adherence) though a behavior skills training paradigm which consists of instructions, modeling, rehearsal, and feedback [37]. Research using this approach could not only assess behavior change in real-time, but also measure potential concurrent changes in impulsivity and self-efficacy to help determine the causal role these psychological constructs play in self-care behavior for individuals diagnosed with either prediabetes or Type 2 diabetes. At present, self-efficacy and other psychological constructs associated with diabetes self-care behaviors (e.g., need for cognition; 18) can be used as markers for potential self-care dysfunction. As noted by Hadj-Abo et al. [18], using these psychological constructs as a screening mechanism for self-care dysfunction may be an efficient way to predict who may need additional diabetes self-care training.

This study is not without limitations. First, the sample size used for the path models is relatively small. The small participant sample may have obscured significant associations between impulsivity, self-efficacy, and self-care behaviors due to a lack of statistical power, especially for participants diagnosed with prediabetes. Second, as mentioned above, obtaining HbA1c levels for an internet-based sample is logistically difficult. However, this biological marker is very important in both the diagnoses and treatment for diabetes. In addition, previous studies have shown an association between DD, HbA1c levels, and some self-care behaviors [11, 12, 14]. Measuring HbA1c levels, DD, BIS-11, and self-efficacy all within the same participants would more accurately demonstrate whether diabetes management is differentially associated with distinct impulsivity types. Lastly, this study was cross-sectional. Thus, casual inferences between impulsivity, self-efficacy, and self-care behaviors is not possible. Future studies measuring the association between these constructs should employ a longitudinal design to facilitate these causal interpretations.

## Conclusions

Impulsiveness based on the BIS-11 was related to several self-care behaviors, with self-efficacy mediating some of the relationships. DD was not related to diabetes self-care adherence in individuals with self-reported Type 2 diabetes or prediabetes diagnoses. Results highlight the need to systematically measure different aspects of multidimensional constructs like impulsivity. This is especially important when trying to determine associations between these constructs and critical health behaviors, that lead to health consequences like Type 2 diabetes. The null results in regard to DD should be interpreted with caution as DD was only assessed for monetary outcomes and the sample size was small for path model analyses. This is the first study to use a choice-based impulsivity measure in an internet-based sample of adults self-reporting a diagnosis of either Type 2 diabetes or prediabetes. Given the prevalence of Type 2 diabetes, this study should be replicated using a longitudinal design with a larger sample to better understand the relationship between impulsivity and diabetes self-care adherence.

## Supporting information

**S1 File. Raw data for all analyzed participants.**
(XLS)

## Author Contributions

**Conceptualization:** Katherine Wainwright, Paul Romanowich.

**Data curation:** Katherine Wainwright, Paul Romanowich.

**Formal analysis:** Katherine Wainwright, Paul Romanowich, Meghan A. Crabtree.

**Methodology:** Katherine Wainwright, Paul Romanowich.

**Project administration:** Katherine Wainwright, Paul Romanowich.

**Resources:** Paul Romanowich.

**Supervision:** Paul Romanowich.

**Visualization:** Paul Romanowich, Meghan A. Crabtree.

**Writing – original draft:** Katherine Wainwright, Paul Romanowich, Meghan A. Crabtree.

**Writing – review & editing:** Katherine Wainwright, Paul Romanowich.

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
