## [Decision Letter · Decision Letter 0]

10 Jun 2021

PONE-D-21-06140

Associations between impulsiveness and self-care adherence in individuals diagnosed with type 2 or prediabetes

PLOS ONE

Dear Dr. Romanowich,

Thank you for submitting your manuscript to PLOS ONE. After careful consideration, we feel that it has merit but does not fully meet PLOS ONE’s publication criteria as it currently stands. Therefore, we invite you to submit a revised version of the manuscript that addresses the points raised during the review process.

We look forward to receiving your revised manuscript.

Kind regards,

Matthew J. Gullo

Academic Editor

PLOS ONE

Additional Editor Comments (if provided):

Concerning Reviewer 1's comments on the BIS-11, psychometric evaluations of the scale tend not to support its subscales and suggest a single factor structure provides a better fit to data (e.g., Steinberg et al., 2013). This could be used to justify your use of a single total score in the path analyses. The construct assessed by BIS-11 is frequently distinguished from that assessed by Delay Discounting. Many researchers view them as assessing different components of impulsivity on theoretical grounds, which may also address one of Reviewer 1's comments (for a review, see Gullo et al., 2014).

Gullo, M. J., Loxton, N. J., and Dawe, S. 2014. “Impulsivity: Four Ways Five Factors Are Not Basic to Addiction.” Addictive Behaviors 39 (11): 1547–56.

Steinberg, L., Sharp, C., Stanford, M. S., and Tharp, A. T. 2013. “New Tricks for an Old Measure: The Development of the Barratt Impulsiveness Scale-Brief (BIS-Brief).” Psychological Assessment 25 (1): 216–26.

Reviewers' comments:

Reviewer's Responses to Questions

**Comments to the Author**

1. Is the manuscript technically sound, and do the data support the conclusions?

Reviewer #1: Partly

Reviewer #2: Yes

2. Has the statistical analysis been performed appropriately and rigorously? 

Reviewer #1: No

Reviewer #2: Yes

3. Have the authors made all data underlying the findings in their manuscript fully available?

Reviewer #1: Yes

Reviewer #2: Yes

4. Is the manuscript presented in an intelligible fashion and written in standard English?

Reviewer #1: Yes

Reviewer #2: Yes

5. Review Comments to the Author

Reviewer #1: Using a cross-sectional design and path analysis, Wainwright et al. examined the relationship between impulsivity, self-efficacy, and self-care adherence and management in adults with a self-reported diagnosis of Type 2 diabetes or prediabetes. The results from this study suggest greater impulsivity may be associated with reduced self-care adherence and that this relationship may be mediated by self-efficacy in participants with Type 2 diabetes. Given diabetes is a chronic condition, that when poorly managed can lead to a variety of serious health complications, investigating mechanisms that can lead to poor management is a relevant addition to this field. While this study highlights several mechanisms that may be important to consider in this context (e.g., impulsivity, self-efficacy), I recommend that this paper not be accepted without major revision.

Overall, this manuscript could be improved with a deeper dive into the research literature to provide concrete justification for the hypotheses and proposed models. In particular, the relationship between impulsivity and self-efficacy was not discussed in enough depth, nor the reason why these variables are important to consider when examining self-care adherence and maintenance in individuals with diabetes or prediabetes. These constructs were not well defined, and it appears some key recent research that would support this study is missing from the Introduction (e.g., Hadj-Abo et al., 2020).

While using advanced statistical analysis (i.e., path analysis) and examining these paths in both Type 2 and prediabetic groups was certainly a strength of this study, the inclusion of certain predictors in these models were not well justified. Given the authors suggested it was important to examine different impulsivity constructs in the Introduction and that the initial correlations suggested not every facet of impulsivity measured by the BIS-11 was associated with the outcome variables, it was unclear why the BIS-11 total score was used as a predictor in the model. Based on the correlations in this study, as well as previous research, it may have been more appropriate to use the non-planning and attention subscales. Additionally, it was also unclear why DD was then included as a predictor given the variable did not correlate with any outcome variables in this study.

While the Discussion section includes a summary of the main findings of this study, the authors could vastly improve this section by including a more in-depth interpretation. For example, Lines 254-258 simply state the study failed to find a relationship between DD and self-care adherence and management problems. While linking these findings to previous research was useful, please include more discussion on why these findings have been so mixed and what the clinical and theoretical implications of this research are. Including clinical implications of all findings from this study in the Discussion would make a valuable addition to the field.

I have included some specific recommendations below:

Abstract

Line 29 – “have never been measured with participants”. This statement is vague and incorrect. Please see Hadj-Abo et al. (2020) who have recently examined these relationships in participants with diabetes.

Line 31 – delete “personality”

Line 27 and throughout – Use the term ‘impulsivity’ instead of ‘impulsiveness’ unless referring specifically to the BIS.

Introduction

Lines 42-45 – As PLOS One is an interdisciplinary journal, consider expanding your definition of diabetes to include a brief discussion on the distinction between Type 1, Type 2, and prediabetes. Based on the information currently available in the introduction, it is not clear why this study has chosen to focus only on Type 2 and prediabetes.

Line 43 – Remove apostrophe from pancreas.

Line 43-44 – Consider including the reference at the end of the sentence to also support the statement of “multiple health complications”. Explicitly listing some of these complications may also help justify the importance of this topic.

Line 47 – Consider a different term for “behavioral treatment”. For example, “behavioral interventions” or “behavioral management strategies” may better capture the actions required to effectively manage diabetes and prevent health complications.

Line 48 – Include reference for “in addition to pharmacological treatments”.

Line 49 – Please expand your definition for “self-care plan complexity”. It is unclear exactly what this term refers to.

Line 51 – Include ‘and’ between “organize execute”

Lines 51-53 – Given that self-efficacy is a core variable in your model, it would be useful to include more detailed discussion on this variable in the Introduction (e.g., be specific about how self-efficacy predicts adherence e.g., does lower or higher self-efficacy predict better adherence? How does lower/higher self-efficacy result in better adherence? What is its relationship with impulsivity?). It would also be helpful to specifically define and discuss diabetes-specific self-efficacy. Reference previous research and also justify why it has been proposed as a mediator between impulsivity and your outcome variables.

Lines 44, Lines 60 etcetera – Inconsistent capitalization of Type 2.

Lines 52-56 – Provide a link between delay discounting and impulsivity so it is clear to the reader why you have introduced DD after mentioning “different impulsiveness constructs” e.g., is DD a behavioural measure of impulsivity? It may also be useful to specifically discuss how DD differs from the facets of impulsivity measured by the BIS-11 and why this distinction is useful.

Line 56-59 – Provide a reference for this statement if available.

Line 61 – Define what HbA1c is for readers who may not be aware of this term.

Lines 60-72 – It would be helpful to provide more discussion as to why DD is associated with some diabetes management behaviours (e.g., non-adherence medication) but not others (e.g., diet adherence). Was this due to methodology, the study sample (e.g., adolescents with Type 1), or particulars of the DD task? It would also be helpful to include studies that have investigated these relationships in participants with prediabetes (if available) and highlight any differences (or expected differences) in findings between these participant groups.

Line 67 – Reword this sentence to state the study examined the relationship between DD and non-adherence and poor glycemic control within a sample of adolescents with Type 1 diabetes (i.e., not between DD and diabetes).

Line 67, 83 etc – Update wording throughout manuscript to ensure person-first language (i.e., adolescents with Type 1 diabetes, not diabetic adolescents). See Dickinson et al. (2017) for guidance on this.

Line 73 – I suggest changing the wording of this sentence to “….is frequently used as a self-report measure of impulsivity”

Lines 73-77 – It would be helpful to mention in this paragraph that BIS-11 includes several subscales measuring different facets of impulsivity. Define these facets and discuss how they may/may not be related to diabetes self-care adherence.

While BMI is a risk factor for diabetes, I believe this section could be much improved by including a summary of previous research specifically investigating the relationship between BIS-11 subscales (and other related measures of impulsivity if applicable) and diabetes related self-care adherence behaviours and outcomes. For example, Hadj-Abo et al., (2020) recently published a study investigating the mediating role of self-efficacy on the relationship between impulsivity and diabetes self-management. This study contains many of the same variables as the current manuscript so the findings from this research would be important to discuss in this Introduction.

Line 82 – Provide a reference for “a similar construct to impulsiveness” and be specific about its relationship with impulsivity. It should be clear why you are discussing this construct in the context of impulsivity, particularly as you have not measured it separately in the current study.

Line 71 & Line 89 – Both studies mentioned in these lines have included an adolescent sample and discussed how the findings are relevant to self-care management for that particular population (i.e., parent observation). Given that research suggests adolescence is a developmental period characterised by increased impulsivity and risk-taking behaviours, it may not be appropriate to extrapolate the findings from these studies to the adult participants used in the current study.

Line 100 – Change “impulsiveness levels” to “impulsivity”.

Materials & Methods

Participants

Line 109 – Provide an explanation as to why only participants over the age of 30 were recruited (particularly as many studies mentioned in the Introduction recruited adolescents)

Line 119 – Demographic Descriptive Table – It would be helpful to split these demographics by participant groups and include a % column.

Measures

-Include the rating scale, scoring information and brief description of psychometric properties (with references) for each measure.

-Capitalize subscale titles e.g., the Control Problem subscale

Line 135 – Suggest rewording “the scale was normalized” to something that better represents what happened (e.g., “As the distribution of the Barrers to Testing Scale was positively skewed, the data was transformed using….”).

Analysis

-Describe what software was used for analyses and reference accordingly (e.g., SPSS v.27; R; MPlus).

-Include mention that a Mann-Whitney U test was also performed on variables that were not normally distributed.

-Given the relationship between impulsivity and BMI/obesity (and the higher rates of BMI in populations with diabetes), it may be appropriate to consider including BMI as a covariate on the path models.

Results

-Specify if there was any missing data and the procedures that were followed to account for it.

Line 171 Table 2

-Include M & SD for BIS-11 total score and range of scores for all measures.

-To assist the reader, report the results from the independent sample t-tests within this table (including the non-significant findings) and then comment on these findings in text. Effect sizes for these tests may also be useful for determining how meaningful the significant differences are.

-Clarify why DCP understanding was reported as N/A in the table.

Line 173 – Refer to table 3 in text before commenting on the results.

Lines 173-176 – Clarify if these relationships were found in both groups separately or if those were combined to get these results. If the groups were combined, please justify this action. Were the relationships the same if the groups were separated?

Table 179 Table 3

-Report all correlations (including non-significant relationships such as BIS-11 & DD, and the relationships between self-efficacy and impulsivity measures) within this table.

-Report relationships between impulsivity measures and self-efficacy and comment on these in text.

Lines 183-197 – Inconsistent use of scale names and constructs measured in text. Use names of specific subscales/measures in text to describe the relationships.

Line 194 – Does this refer to the total BIS score? Report these correlations in the table.

Lines 198-205 – As stated earlier, include results of t tests in Table 3 and comment on these tests in text. Include statistics for non-sig. relationships

Path Models

Lines 206-251 - Given the aim of this study was to examine the relationships between different impulsivity constructs, and that different correlation relationships between BIS-11 subscales and self-care/management problems were found, it is unclear why a total BIS-11 score (rather than subscales, specifically attention & non-planning) was used in the path model. In addition, as DD was not correlated with any model variables in this study it is unclear why this variable was included in the path models as a predictor of these outcomes. It may be more appropriate to consider using simple mediation models with the variables that have demonstrated significant relationships.

I am also concerned that the sample size for each group may be too small to test a path model of this size. It is typically recommended to have a minimum 20 participants per parameter for SEM. While it may be a difficult population to recruit, it is important to highlight this limitation in the Discussion or use a different type of analysis.

Line 221 – It is inappropriate to call a non-significant relationship “marginally significant”. Remove this phrase from the manuscript (including from the Abstract and Discussion) and provide a correct interpretation of these relationships.

Discussion

-There is a strong sense of repetition of the results section within this Discussion. While providing a summary of the results is important, please also include an interpretation of these findings in the context of previous research and the population sampled. Including a clinical implications section would be a valuable addition.

-Include a discussion on how the findings from this study supported/did not support the hypotheses.

-Please include a clear limitations and future directions section.

Line 280-281 – Remove text in brackets.

Line 284 – Use third-person language. Check the rest of manuscript for this language as well.

Lines 303-307 – It is unclear why this paragraph has been included here. Is this paragraph being used to explain why there are often mixed findings when studies have used DD with populations with diabetes? Please provide some clarity and supporting references, and an explanation as to why other commodities (e.g., food) may be more appropriate.

Lines 312-313 – “an open question” is quite vague. Please clarify and expand the statement about causal status of self-efficacy. How does this study contribute to this status?

Lines 316-317 – These relationships were also not found in participants with prediabetes.

Line 319 – type 2 diabetes is not a health behaviour.

Lines 319-320 – Provide a brief reason why DD results should be interpreted with caution as this was not immediately clear from the discussion.

Line 320 – Delete “relatively robust”. This is term is quite vague and it is unclear what is meant by this statement.

Line 322 – Rather than just suggesting replication, provide a more specific example e.g., longitudinal evaluation in larger sample etc?

Suggested References

Dickinson, J. K., Guzman, S. J., Maryniuk, M. D., O’Brian, C. A., Kadohiro, J. K., Jackson, R. A., ... & Funnell, M. M. (2017). The use of language in diabetes care and education. The Diabetes Educator, 43(6), 551-564.

Hadj-Abo, A., Enge, S., Rose, J., Kunte, H., & Fleischhauer, M. (2020). Individual differences in impulsivity and need for cognition as potential risk or resilience factors of diabetes self-management and glycemic control. PloS one, 15(1), e0227995

Reviewer #2: Thanks for the opportunity to review this intetesting paper on the relationships between impulsiveness and self-care in tYPE II zdiabetes and prediabetes. Results are intetesting and the paper is overall well written, as well as statisical Analyses are well conducted.

I have only minor suggestione to make.

Specifically, in my opinion the introduction in too long. Please try to reduce it. For example, you could omit the general paragraphs on diabetes and prediabetes. In my opinion, the are well known concepts. However, I'll briefly cite some paper examining the relationship between diabetes and psychopathology (especially the relevance on specific personality disorders in individuale with diabetes or brutale diabetes). Some of these personality disorders are characterized by impulsiveness and nay probably influence the brittleness of diabetes. For details, you could cite PMID: 27524279. Also alexithymia could interfere in interpersonal relationships and partially influence impulsiveness in isubjects with instabile diabete (PMID: 31125008).

These concepts may be also considered in the discussione of your results, especially in relation with the link you found between impulsiveness and self-care.

6. PLOS authors have the option to publish the peer review history of their article (what does this mean?). If published, this will include your full peer review and any attached files.

Reviewer #1: No

Reviewer #2: No

---

## [Author Response · Author response to Decision Letter 0]

20 Aug 2021

The Response to Reviewers file has been uploaded. See attached file for detailed response to all reviewer comments/suggestions.

---

## [Decision Letter · Decision Letter 1]

4 Nov 2021

PONE-D-21-06140R1Associations between impulsivity and self-care adherence in individuals diagnosed with type 2 or prediabetesPLOS ONE

Dear Dr. Romanowich,

Thank you for submitting your manuscript to PLOS ONE. After careful consideration, we feel that it has merit but does not fully meet PLOS ONE’s publication criteria as it currently stands. Therefore, we invite you to submit a revised version of the manuscript that addresses the points raised during the review process.

Both Reviewers agree that significant progress has been made in addressing their initial comments. Some additional matters require attending to, especially those outlined by Reviewer 1, and I encourage the authors to address these in a revised submission.

We look forward to receiving your revised manuscript.

Kind regards,

Matthew J. Gullo

Academic Editor

PLOS ONE

Reviewers' comments:

Reviewer's Responses to Questions

**Comments to the Author**

1. If the authors have adequately addressed your comments raised in a previous round of review and you feel that this manuscript is now acceptable for publication, you may indicate that here to bypass the “Comments to the Author” section, enter your conflict of interest statement in the “Confidential to Editor” section, and submit your "Accept" recommendation.

Reviewer #1: (No Response)

Reviewer #2: All comments have been addressed

2. Is the manuscript technically sound, and do the data support the conclusions?

Reviewer #1: Partly

Reviewer #2: Yes

3. Has the statistical analysis been performed appropriately and rigorously? 

Reviewer #1: No

Reviewer #2: Yes

4. Have the authors made all data underlying the findings in their manuscript fully available?

Reviewer #1: Yes

Reviewer #2: Yes

5. Is the manuscript presented in an intelligible fashion and written in standard English?

Reviewer #1: Yes

Reviewer #2: Yes

6. Review Comments to the Author

Reviewer #1: The authors have done an excellent job so far addressing the comments on their first draft - well done! However, there are still a few areas of improvement that require addressing before this article should be accepted for publication. Please find these listed below:

Abstract:

- Lines 27-30: This is a very long sentence and suggests that both impulsivity measures have not been measured in the same study before (regardless of participant sample). This is not the case (when referring to participant samples beyond diabetes). I would suggest deleting “participants, or for” and specifying that delay discounting has not been measured in individuals with Type 2, while neither delay discounting & BIS have been measured in pre-diabetes (if that is the case).

- To clarify your findings, It would be useful to also mention in the abstract that the choice-based personality measure was not associated with outcome variables so as to really support your final sentence in the abstract (that “ how specific impulsivity constructs ….”)

- Lines 37-39: The final sentence of this abstract is somewhat misleading. Your study highlights what specific impulsivity constructs may be more useful to target in interventions but not how these constructs can be incorporated into interventions. I recommend rewording this sentence so it more accurately reflects your study.

Introduction:

- Line 56: Move the word “can” to after “requirements that”

- Line 69: Insert commas as such: “impulsivity, such as a lack of inhibition, also”

- Line 70: Include a comma after the last bracket

- Line 112, include a comma after references

- Lines 58-73: I think the authors have done a wonderful job expanding on the definition of self-efficacy and how this relates to self-care behaviours.

- Line 95-96: I think it might be helpful to include a brief overall summary about what the consistent findings mean in relation to self-care, so it is very clear to the reader. Something like “Overall, the findings of these studies suggest that higher levels of impulsivity, as measured with the DD task, are associated with poorer diabetes self-care adherence. However, the findings…..”

- Lines 98-101: Great job including a summary of the BIS-11 subscales and how they are likely linked to self-care adherence.

- Line 103: specific how BIS-11 was related to BMI e.g., positive, negative relationship (just so it is very clear to the reader).

- Line 112: insert comma after references.

- Line 116: Due to a run on sentence, delete “like other important heath behaviours”. If you want to keep this in the introduction, I suggest including this comment as a separate sentence.

- Line 117: insert comma after “studies”

- Line 123: replace “for” with “in”

Materials & Methods:

- Line 196: Reword sentence to “Some items were reversed scored.”

- Line 208: Are the authors referring to internal consistency and a valid measure of impulsivity here? Please report alpha values (or equivalent) are per previous measures in this manuscript, and specify what is meant by ‘valid’

Results:

- Please ensure your Tables align with PLOS formatting requirements.

- There is still inconsistent use of scale/subscale/measures in text/tables. E.g., see table 2, Lines 241-245, Lines 348-249.

- Lines 259-273 and Table 3. This section should come before correlations and path models as it provides further justification as to why you have chosen to analyse these groups separately. Table 3 should be introduced in text prior to appearing.

- Table 3: Reconsider the layout of the final column in this table. Not all rows in this table at t-tests, and the header of this column does not represent every value reported underneath it.

- Table 2: Please include the total BIS-11 score in this correlation table and discuss in text.

- Table 2 and Lines 227-256: Based on the authors’ justification in Lines 220-222, it is unclear why the BIS-11 subscale correlations are reported and total score has been omitted here. The lack of relationship between some subscales of this measure for Type 2 (as reported in Line 231) and the differential relationships between these subscales and self-care adherence and self-efficacy appear at odds with this justification. As a reviewer, I am still not convinced that it is appropriate to use the total BIS-11 score in the path analyses. Theoretically, it makes sense as to why significant negative correlations are noted between BIS11 Non-Planning/Attention and these outcomes. Please ensure adequate and clear justification for this decision process is provided, beyond just the one study suggesting a single factor structure.

- Since you have now included BMI as a covariate, please include this in your correlations and t tests and provide some justification for this decision (e.g., based on previous research, significant correlations in the current study etc).

Path Models

- Capitalise Type 2 in Line 279 and check the remaining manuscript for consistency.

- Lines 283 & 287: For consistency, report “controlling for BMI” the same across these figure notes.

- Lines 280 & 289: Delete ^p < .10 as this is not used in the figures.

- Please provide justification for including BMI as a covariate as this appears to be a new addition since the first draft. From your results, it appears that BMI is not correlated with your outcome variables, so again, it is unclear why this covariate was included. Are the path models the same if you do not include BMI as a covariate? If so, report the path models without the covariate and include a note that the same models were run with BMI as a covariate but findings did not differ (if justification for BMI covariate is included).

Discussion

- To reorientate the readers, it is helpful to restate the hypotheses at the beginning of the discussion. Include a brief statement of the hypotheses following the first sentence in the discussion. This should be the first paragraph, then flow onto the findings. Please refer to PLOS guidelines on writing and formatting the Discussion and Conclusion sections of this manuscript - https://plos.org/resource/how-to-write-conclusions/

- Use third-person language (e.g., Lines 332, 336) and check the remaining manuscript for this.

- Line 328: replace ‘predictive’ with ‘correlated’ to more accurately represent the data analysis and findings.

- Lines 328-331: Discussion about DD might be better suited if it were included with the content in Lines 378-406

- Line 335 – replace ‘this’ for ‘the’

- Lines 332-346: Overall, this paragraph would benefit from a re-write. The discussion bounces between the findings of Hadi-Abo and the current study’s results and hypotheses which makes it somewhat difficult to follow. I would suggest this type of structure: State the results. State how the results support/do not support the hypotheses. State how this relates to previous research. Explanation of findings in the context of the hypotheses and previous research. Be careful to avoid run-on sentences and ensure appropriate grammar (particularly the use of commas).

- Line 344-346: This concluding sentence is too casual (i.e., replace “Perhaps there are” with “There may be”) and does not provide an explanation as to why self-efficacy did not mediate. It might be useful to hypothesise why this result occurred based on your knowledge of the literature, the constructs, and this study’s findings.

- Lines 347-348: This sentence requires a slight re-write as it contains grammatical and word choice errors. The connection between this sentence and the current study’s results are unclear, please provide the link so it is clear to the readers. Something along the lines as “The current study aimed to address this gap in the literature…” would likely suffice.

- Lines 347-363: Similar to the feedback provided about Lines 332-346, this paragraph would also benefit from a re-write. At times, it was unclear if the authors were discussing the results related the pre-diabetes, or Type 2, or both (e.g., Lines 354-357)

- Line 360: Please clarify what the authors mean by “not high enough to impact…”. By scoring lower on self-efficacy, this would imply less adherence/self-care (at least based on Type 2 findings). It is unclear why scoring lower on this measure would result in a non-significant relationship. The significant difference between self-efficacy between participant groups implies that Type 2 have higher levels of self-efficacy compared to prediabetes – is this because they have likely spent more time engaging in these behaviours in the past? This would be important to explore in the discussion as it is likely an important target for clinical intervention (at least for Type 2)

- Line 367: As stated in the previous feedback, please remove the term “marginally significant” and provide the correct interpretation of results.

- Line 337: Remove impulsivity. By the definition of personality, impulsivity is unlikely to be a treatment-modifiable factor. Given impulsivity is the same across these participant groups, longitudinal research should focus on factors that are likely to change over time and with treatment (such as self-efficacy and adherence).

- Lines 384-387: This is a run-on sentence. Consider rephrasing and writing in more succinct sentences.

- Line 390: Delete “To date, no study has done this.”

- There are several errors in punctuation, grammar, and sentence/paragraph structure throughout the discussion and other sections of this manuscript. While I have highlighted somes of these errors, it is recommended that the authors carefully review and copyedit this manuscript before submitting the revised draft. This resource may be useful https://plos.org/resource/how-to-edit-your-work/

Reviewer #2: I have no other comment. In my opinion, the manuscript is now ready for publication in PLoSOne.

Only a comment on your answer to my previous suggestion: "...The articles on brittleness of diabetes only focused on those patients diagnosed with Type 1 diabetes. Therefore, we are unsure whether this addition will aid in an understanding of relationships between impulsiveness and Type 2/prediabetes. This information may fit in the clinical implications section where we outline the role psychological constructs may play in remediating self-care adherence. However, this still feels somewhat ad hoc. For these reasons, we have chosen not to include brittleness in the current manuscript revision".

In my opinion, it is not important whether patients were diagnosed with Type I or TYpe II diabetes/prediabetes. I think that brittleness may be due to a bad self-care adherence related to impulsiveness and specific Cluster B personality traits. In studies on brittleness on Type I diabetes it has been shown that there was a relationship between glycaemic instability and these specific cluster B personality traits. I think that the same psychopathological substrate is what you have observed in patients with type II diabetes.

7. PLOS authors have the option to publish the peer review history of their article (what does this mean?). If published, this will include your full peer review and any attached files.

Reviewer #1: No

Reviewer #2: **Yes: **Lorenzo Pelizza

---

## [Editor Report · Decision Letter 2]

2 Feb 2022

Associations between impulsivity and self-care adherence in individuals diagnosed with type 2 or prediabetes

PONE-D-21-06140R2

Dear Dr. Romanowich,

We’re pleased to inform you that your manuscript has been judged scientifically suitable for publication and will be formally accepted for publication once it meets all outstanding technical requirements. There is also a minor error that will need to be corrected during copyediting (see below).

Kind regards,

Matthew J. Gullo

Academic Editor

PLOS ONE

Additional Editor Comments:

Please correct this minor error during copyediting: On Line 284-285: Correct the text to specify that two relationships changed when not controlling for BMI (not one).
---

## [Editor Report · Acceptance letter]

1 Mar 2022

PONE-D-21-06140R2 

Associations between impulsivity and self-care adherence in individuals diagnosed with Type 2 or prediabetes 

Dear Dr. Romanowich:

I'm pleased to inform you that your manuscript has been deemed suitable for publication in PLOS ONE. Congratulations! Your manuscript is now with our production department. 

Kind regards, 

on behalf of

Assoc. Prof. Matthew J. Gullo 

Academic Editor

PLOS ONE